# Obesity Subtyping: The Etiology, Prevention, and Management of Acquired versus Inherited Obese Phenotypes

**DOI:** 10.3390/nu14112286

**Published:** 2022-05-30

**Authors:** Edward Archer, Carl J. Lavie

**Affiliations:** 1Research & Development, EvolvingFX, LLC., Fort Wayne, IN 46835, USA; 2Department of Cardiovascular Diseases, John Ochsner Heart & Vascular Institute, Ochsner Clinical School, The University of Queensland School of Medicine, New Orleans, LA 70121, USA; clavie@ochsner.org

**Keywords:** inherited, acquired, obesity, diet, exercise

## Abstract

The etiology of obesity is complex and idiosyncratic—with inherited, behavioral, and environmental factors determining the age and rate at which excessive adiposity develops. Moreover, the etiologic status of an obese phenotype (how and when it developed initially) strongly influences both the short-term response to intervention and long-term health trajectories. Nevertheless, current management strategies tend to be ‘one-size-fits-all’ protocols that fail to anticipate the heterogeneity of response generated by the etiologic status of each individual’s phenotype. As a result, the efficacy of current lifestyle approaches varies from ineffective and potentially detrimental, to clinically successful; therefore, we posit that effective management strategies necessitate a personalized approach that incorporates the subtyping of obese phenotypes. Research shows that there are two broad etiologic subtypes: ‘acquired’ and ‘inherited’. Acquired obesity denotes the development of excessive adiposity after puberty—and because the genesis of this subtype is behavioral, it is amenable to interventions based on diet and exercise. Conversely, inherited obesity subsumes all forms of excessive adiposity that are present at birth and develop prior to pubescence (pediatric and childhood). As the inherited phenotype is engendered in utero, this subtype has irreversible structural (anatomic) and physiologic (metabolic) perturbations that are not susceptible to intervention. As such, the most realizable outcome for many individuals with an inherited subtype will be a ‘fit but fat’ phenotype. Given that etiologic subtype strongly influences the effects of intervention and successful health management, the purpose of this ‘perspective’ article is to provide a concise overview of the differential development of acquired versus inherited obesity and offer insight into subtype-specific management.

## 1. Introduction

Obesity is a global health concern [1,2,3,4,5], with the prevalence in the U.S. exceeding 40% in adults and nearly 20% in children and adolescents [6]. Although efforts to stem the increasing prevalence have been unsuccessful, research has led to a clearer understanding of its etiology and how obesity impacts cardiometabolic diseases, such as type-2 diabetes mellitus (T2DM), dyslipidemia, cardiovascular disease, and hypertension [7,8].

Despite these conceptual advances, the development of effective prevention and management protocols has been less successful. Although lifestyle modifications are the cornerstone of obesity management, few individuals achieve long-term benefits with ‘one-size-fits-all’ diet and exercise approaches [9]. We posit that this lack of success is not due to a deficiency of willpower or adherence by participants and patients but is engendered by the failure to recognize that the obese phenotype is not a single homogenous condition [10]. To be precise, obesity, despite being an increasingly common phenomenon, has a complex, idiosyncratic etiology—with inherited, behavioral, and environmental factors determining the age and rate at which excessive adiposity and cardiometabolic diseases develop. 

Thus, because research suggests that that the etiology of an obese phenotype (how and when it developed initially) strongly influences the short-term effectiveness and long-term outcomes of lifestyle interventions [1,3,11,12], successful obesity management necessitates the subtyping of phenotypes. As such, the purpose of this ‘perspective’ article is to provide a concise overview of the differential development of acquired versus inherited obese phenotypes and offer insight into subtype-specific obesity prevention and management. 

## 2. Etiologic Subtypes of Obesity

Although the defining characteristic of obesity is an excess of bodyfat [2,12], the age and rate at which excessive adiposity develops vary as a result of inherited, behavioral, and environmental factors. Thus, because the obese phenotype may be engendered at any point in an individual’s development—from the prenatal period to senescence—research suggests two broad etiologic subtypes: ‘acquired’ and ‘inherited’ [1,3]. 

Acquired obesity, also known as ‘adult-onset’, denotes the disproportionate development of adiposity after puberty. The genesis of this phenotypic subtype is essentially behavioral, with physical activity (PA) and subsequent hyperphagia (overconsumption) being the major determinants. Stated simply, ‘moving too little’ leads to ‘eating too much’ and together these pathologic behaviors lead to acquired obesity and cardiometabolic diseases, such as T2DM [1]. 

Conversely, inherited obesity—also known as pediatric or childhood—subsumes all forms of excessive adiposity that are present at birth or develop prior to pubescence. Inherited obesity can be further subdivided into ‘non-genetic’ (common) and ‘genetic’ (rare) obesity. Common inherited obesity is a ubiquitous, complex, *quantitative* (continuously distributed) phenotype characterized by altered fat, muscle, and pancreatic beta-cell development and function. These structural (anatomic) and physiologic (metabolic) alterations are engendered during prenatal development, and as such, are largely irreversible [3,11,13]. 

Genetic obesity refers to Mendelian disorders that result in discrete, *qualitative* phenotypes that display excessive adiposity (e.g., leptin deficiency, Prader-Willi syndrome). As explained in detail below, because the genesis of common inherited obese phenotypes differs considerably from the genesis of the genetic inherited phenotype, it is important to distinguish between these subtypes in prevention, diagnosis, and management. As genetic obesity is rare [4], and the role of ‘genes’ in common obesity is limited (see Section 3 below), in this review, we limit our discussion to common (nongenetic) inherited forms of obesity.

## 3. Nongenetic versus Genetic Inheritance and the Role of Genes in Obesity

We have written extensively on the role of nongenetic inheritance in the development of obesity and T2DM, and how the conflation of the term ‘inherited’ with ‘genetic’ has led to confusion [3,14,15]. Given that a detailed exposition of the conceptual and empirical foundation for our work is beyond the scope of this article, we offer a concise overview below and direct our readers to select publications [1,3,12,13,14,15,16].

Briefly, the functional unit in biology and biological inheritance is the cell, and because each cell’s idiosyncratic nature and spaciotemporal context determines gene expression, it is important to distinguish between nongenetic (cellular) inheritance and the two forms of genetic inheritance (nuclear and mitochondrial) [3,17,18,19]. For example, the fundamental difference between monozygotic (identical) and dizygotic (fraternal) twins is inherent in the nomenclature—identical twins develop from a single cell (a fertilized egg) whereas fraternal twins develop from two different cells (two fertilized eggs). Thus, fraternal twins differ in both cellular and genetic inheritance whereas identical twins do not. Therefore, the greater phenotypic disparity displayed by fraternal twins is due to differences in the genotypic expression induced by different cells in concert with inter-twin differences in genotype. Yet despite the variability in the developmental competence of any given population of eggs, the functional distinction between cellular and genetic inheritance is ignored routinely by those who infer genetic causality from ‘twin-studies’ and heritability statistics.

To be precise, our work demonstrated that “*there are no ‘genes for’ quantitative (i.e., non-discrete) phenotypes, such as common obesity and metabolic diseases (e.g., T2DM)*.” [1]. We further detailed the “*fatal flaws of twin studies*” and showed why “*estimates of genetic heritability are misused, and often meaningless statistical abstractions derived from attempts to impose an artificial and false dichotomy (i.e., nature* vs. *nurture* (genes vs. environment)) *on demonstrably non-dichotomous biologic processes*” [1].

These conclusions—which form the basis for our perspective on the limited role of ‘genes’ in obesity—are most clearly supported by the nonlinear processes that lead to ‘one-to-many’, ‘many-to-one’, and ‘many-to-many’ genotype–phenotype relations. These processes include reaction norms, phenotypic accommodation, alternative splicing, RNA editing, chimeric transcripts, protein multifunctionality, epistatic variance, maternal effects, the metabolic regulation of transcription, and post-translational modifications. 

Thus, a ‘great deal of biology’—both established and undiscovered—links an individual’s genotype, the cellular expression of that genotype, and the development of specific phenotypes; therefore, as Felder and Lewontin wrote, there is *“a vast loss of information in going from a complex machine* [an organism] *to a few descriptive parameters* [heritability estimates]*”* [20]. Moreover, because estimates of genetic heritability are mere statistical associations, they cannot be used to quantify the relative contributions of presumed etiologic factors outside of highly controlled animal and plant breeding operations [1,21]. In other words, ‘correlation does not equal causation’—especially when the relations are nonlinear, and the fundamental constructs are inherently flawed or misconstrued.

In sum, the emerging field of non-genetic inheritance [17,18,19] and our work suggests that genes are *“tools of the cell”*, and as such, “*are merely a necessary but not sufficient component for the development of obesity/T2DM phenotypes*” [14]; therefore, an understanding of the etiology, prevention, management, and treatment of these phenotypes *“will not be found in the genome”* [3].

As detailed below, because the etiologies of acquired and ‘non-genetic’ (common) inherited obese phenotypes differ markedly, strategies for their prevention and management must be subtype-specific. 

## 4. Acquired Obesity: Its Etiology and Response to Intervention

Although the etiology of acquired obesity is often contested [1], there is strong evidence dating from the mid-20th century that reductions in PA, high physical inactivity (PI), and excessive sedentary behavior (SB) are strong determinants of this phenotype in both human and non-human animals [1,12,22,23,24,25,26,27,28,29,30,31,32,33]. To summarize briefly, first, PA is the major *modifiable* determinant of caloric consumption [27,28,31,34,35,36,37]. Second, when individuals reduce their PA, their consumption declines more slowly than their caloric expenditure [27,28,31,34,35,36]. This leads to relative hyperphagia (overconsumption) and positive energy balance—with individuals consuming more calories than they expend. 

Third, as PA declines, the energetic demands of skeletal muscle decline. This reduces the number of calories partitioned to skeletal-muscle and increases the number of calories available for storage in fat-cells (adipogenic partitioning). Fourth, PI decreases skeletal muscle insulin-sensitivity, which induces hyperinsulinemia (higher levels of insulin) during and after each meal with concomitant increments in adipogenic partitioning and decrements in lipolysis. The confluence of PI-induced hyperphagia and hyperinsulinemia causes a greater percentage of the calories consumed at each meal to be stored and sequestered in fat cells (reduced lipid turnover) with concomitant increments in body and fat mass. 

When PI and excessive SB become habitual, the attendant metabolic perturbations [33] lead to acquired obesity via increments in fat-cell size, number (hypertrophy and hyperplasia, respectively), and ectopic development (fat-cell intrusions into non-adipose tissue). If the increased demands for insulin production and caloric storage cannot be met by parallel increments in pancreatic beta-cell functioning and fat-cell plasticity, the declining skeletal muscle insulin-sensitivity progresses to whole-body insulin-resistance, and over time, to overt T2DM [38,39,40,41]. Evidence for these phenomena was established decades ago, with the loss of skeletal muscle insulin sensitivity being the initial and primary metabolic insult in cardiometabolic diseases [38,39,40,41]. Thus, PI, high levels of SB, and concomitant hyperphagia are the major etiologic factors leading to acquired obesity and cardiometabolic diseases [1,12,16,30,33,38].

Nevertheless, despite the strong influence of PA on the development of acquired obesity and T2DM, the management of these metabolic maladies *must* include dietary interventions because exercise-only interventions have trivial impacts on body mass and weight loss, despite clinically important impacts on body composition, and blood glucose and insulin levels.

## 5. The Prevention and Management of the Acquired Obese Phenotype

As the genesis and maintenance of the acquired obese phenotype are largely behavioral (moving less and eating more), prevention entails adequate levels of daily PA and relative caloric consumption from childhood to senescence. To be precise, 30–60 min of daily PA and a physical activity level (PAL) reaching 1.7–1.8, are necessary for the primary prevention of acquired obesity and the maintenance of a reduced (post-obese) phenotype [31,42,43]. Furthermore, in the early stages of development, the acquired subtype is extremely amenable to interventions emphasizing diet, PA, and exercise; however, it is important to note that despite the demonstrated impact on metabolic (glycemic and lipidemic) control, exercise-only interventions have a limited impact on weight-loss and body mass [44,45]. Thus, dietary and caloric restriction *must* play a dominant role if body mass is to be reduced. 

This may be particularly important in patients with obesity and cardiometabolic diseases, such as dyslipidemia, especially hypertriglyceridemia, hypertension, and elevated blood glucose levels, including metabolic syndrome and T2DM. These patients require increased PA and exercise in concert with reductions in caloric intakes, particularly simple and complex carbohydrates and alcohol—even more so than reductions in fat intake—to improve both weight and metabolic control [46,47,48].

It is important to note, however, that if the chronic positive energy balance and metabolic perturbations induced by PI and excessive SB continue over time, the growth in the number of fat-cells (hyperplasia), in concert with the degradation of pancreatic beta-cell function and insulin sensitivity eventually lead to diminished health and responsiveness to lifestyle interventions. As such, long-standing acquired obesity will resemble the common inherited obese phenotype in its response to intervention [1,3,12].

## 6. Inherited Obesity: Its Etiology and Response to Intervention

In contrast to the behavioral genesis of the acquired (adult-onset) phenotype, the common inherited phenotype is engendered during in utero (prenatal) development. As briefly explained below, and detailed elsewhere [3,11,12,13,14,15], this subtype exhibits irreversible structural (anatomic) and physiologic (metabolic) perturbations engendered by the mother’s behavioral and metabolic phenotypes (e.g., PA levels, adiposity, glycemic control).

Briefly, it is well-established that during pregnancy, a mother’s cells compete for calories with those of her fetus [3,12,13,16]. Thus, to ensure that the fetus receives the number of calories it needs for development, pregnancy leads to hormonal changes that induce insulin-resistance in maternal skeletal muscle. This naturally developing insulin-resistance increases caloric consumption while decreasing the number of calories partitioned to maternal skeletal muscle. This leads to increased maternal serum lipid and glucose levels with concomitant increments in maternal body and fat mass, and caloric transfer to the fetus [3].

For comparison, stunting and common inherited obesity represent opposing ends of the maternal–fetal competitive continuum and they impact at least three generations: the mother, the fetus, and the germline of female fetuses. Stunting develops when a mother’s diet and body-fat stores cannot keep pace with the competitive demands of her cells and fetal development. This causes fewer fetal muscle, fat, bone, and pancreatic beta-cells to be created, and permanently alters the offspring’s structural (anatomic) and physiologic (metabolic) phenotypes (e.g., shorter height and impaired glucose and lipid metabolism). These changes are irreversible and substantially increase the risk of cardiometabolic diseases [49,50,51].

Conversely, common inherited obesity is engendered by insufficient maternal PA and metabolic control which reduces the competition for calories between mother and fetus. More specifically, when the naturally occurring insulin-resistance of pregnancy acts in concert with the pathological insulin-resistance induced by maternal PI and excessive SB, the escalation in insulin-resistance exponentially increases caloric consumption, while decreasing the number of calories partitioned to maternal skeletal muscle. This causes an excessive number of calories to be transferred to the fetus—which stimulates a disproportionate increment in fat-cell size and number, fetal insulin production, and dysfunctional skeletal muscle development (more structural and less contractile elements) [3,11,12,13,14,15].

These pathologic ‘maternal-effects’ (non-genetic mechanisms of inheritance) are irreversible and produce children who are predisposed to *‘eating more and moving less’*, independent of genotype [3,11,12,13,14,15]. Infants and children with this subtype will consume more calories than those with normal phenotypes because their excessive fat-cell hyperplasia, reduced skeletal muscle function, and hyperinsulinemia, increase the number of calories stored and sequestered in fat-cells after each meal—both in adipose tissue and ectopically. Over time, this adipogenic partitioning causes increments in body and fat mass, and concomitant obesity [1,12,16]. These ‘maternal-effects’ offer a comprehensive explanation for the inheritance of compromised metabolic phenotypes in both human and nonhuman animals [3,12,13,14,15].

Thus, increments in childhood obesity and adolescent T2DM are most plausibly explained by the substantial decline in PA and increments in SB over the past 50 years by young women and mothers [52,53,54]. As the PI-driven maternal-effects escalated from one generation to the next, the prevalence of both obesity and T2DM increased markedly [3,11,12,13,14,15]. Our research suggests that these pathological maternal effects also explain the increased prevalence of obesity and cardiometabolic maladies in nonhuman mammals inclusive of dogs, cats, laboratory mice, monkeys, and feral moose [1,12]. 

## 7. The Prevention and Management of the Inherited Obese Phenotype

The inherited obese phenotype represents a continuum of metabolic perturbations instantiated during prenatal development. Thus, unlike acquired obesity, the structural (anatomic) and physiologic (metabolic) perturbations are not a behavioral manifestation, but are inherent to the phenotype, and therefore, are irreversible. This means that the prevention of common inherited obesity must begin with the current generation of female children and adolescents (future mothers). Sufficient increments in pre-pubertal, pubertal, pre-conception, and prenatal PA will ameliorate or prevent the pathologic maternal effects that lead to this phenotypic subtype. More specifically, as with the prevention of acquired obesity, future mothers must perform at least 30–60 min of daily PA and reach a PAL of 1.7–1.8 to prevent the development of common inherited obesity in future generations.

Nevertheless, once instantiated in utero, the structural and physiologic perturbations engendered by accumulative maternal effects are irreversible. To be precise, the inherited phenotype exhibits both hypertrophic and hyperplastic obesity (greater fat-cell size and number) in concert with dysfunctional pancreatic-cell function and reduced muscle-cell contractility. No behavioral interventions can reduce the number of fat-cells, nor wholly overcome the reduced muscle-cell function; therefore, individuals with this subtype will always find it more difficult to ‘move more and eat less’ than individuals with normal or acquired obese phenotypes.

Importantly, as detailed in the following section, the amount of PA and caloric restriction necessary to induce and maintain weight loss may be beyond many individuals’ physical and/or psychological capacity for exercise and caloric deprivation. Thus, the inherited obese phenotype is less amenable to interventions than the acquired subtype and in many cases, the best health trajectory achievable will be a ‘fit but fat’ phenotype [7,55,56,57,58,59].

## 8. The *‘Metabolic Tipping-Point’* and Its Effect on Intervention

There is a large body of observational and experimental research dating from the 1950s showing that although body mass and composition and concomitant basal energy expenditure are the major determinants of caloric consumption [60,61,62,63,64], PA is the major *modifiable* determinant of consumption, expenditure, and storage [16,27,28,31,34,35,36,37,65,66,67,68]. Thus, because PA plays an essential role in all aspects of metabolism, we previously coined the term *‘Metabolic Tipping-point’* to denote the amount of PA necessary to prevent overconsumption and weight gain [1,12]. As briefly explained below, and in detail elsewhere [1,12], this concept offers a concise framework for understanding the heterogeneity of response of caloric consumption and body and fat mass to altered levels of PA.

As depicted in Figure 1, PA, body mass, and caloric consumption have complex, nonlinear relations [16]. When an individual’s PA declines below their lower metabolic tipping-point (the left side of Figure 1), caloric intake declines more slowly than energy expenditure (a nonlinear relation). This leads to increments in body fat and mass, and decrements in skeletal muscle insulin-sensitivity. If habitual, these individuals will develop acquired obesity and T2DM—dependent on fat-cell plasticity and pancreatic beta-cell function. Nevertheless, any intervention that increases their PA above their lower metabolic tipping-point will reduce hyperphagia, positive energy balance, and prevent further gains in body and fat mass. Nevertheless, as explained in a previous section, *caloric restriction is essential* if the excess body mass is to be reduced because interventions that rely exclusively on PA and exercise have trivial effects on body mass.

When individuals maintain PA levels between the upper and lower metabolic tipping-points (the center portion of Figure 1), their body and fat mass remain stable, regardless of increments and decrements in PA within this range. This occurs because of a linear relation between caloric consumption and expenditure at moderate levels of PA. Thus, as PA increases, caloric consumption increases in parallel. To be precise, the *nonlinear* relations below the lower tipping-point explains why decrements in PA lead to increments in body and fat mass in highly sedentary individuals, whereas the *linear* relation between PA and consumption in the range between the upper and lower metabolic tipping-points explains why increased PA and exercise have little or no effect on body mass in individuals who are already moderately active.

Conversely, when individuals increase their PA above their upper metabolic tipping-point (the right side of Figure 1), they experience declines in caloric consumption, basal energy metabolism, energy expenditure, and body and lean mass. This level of PA is not sustainable and leads to incomplete recovery, reduced physical performance, injury, and exhaustion [70].

In summary, the left side of Figure 1 depicts the *nonlinear* relations between caloric consumption and expenditure, and the concomitant development of acquired obesity and T2DM. The center panel depicts the *linear* relations between PA and caloric consumption and explains why exercise interventions without caloric restriction will not reduce body and fat mass. The right side of Figure 1 depicts unsustainable levels of PA that lead to the loss of body and lean mass. Therefore, it is the transition from a nonlinear to a linear relation between caloric consumption and expenditure as PA increases from a sedentary to an active lifestyle that explains the heterogeneity of response to diet and exercise in individuals with varied levels of baseline PA.

Nevertheless, what Figure 1 does not depict is how an individual’s obese subtype impacts their metabolic tipping-points. Because the acquired obese phenotype is essentially a behavioral phenomenon, any intervention that increases PA and reduces caloric consumption will be successful in the early and mid-stages of phenotypic development; however, the longer the physical inactivity-induced metabolic perturbations continue, the less amenable to intervention the acquired subtype becomes. In this respect, long-standing acquired obesity will mimic the inherited subtype in its response.

Conversely, individuals with an inherited subtype represent a continuum of irreversible structural (anatomic) and physiologic (metabolic) perturbations that are inherent to their phenotype. As such, the amount of PA and caloric restriction necessary to reduce body mass and maintain weight loss depends on where they fall in the continuum of perturbations—from mild to extreme. The more extreme an individual’s inherited obese phenotype, the higher their metabolic tipping points, and the greater the amount of PA and caloric restriction required to prevent overconsumption and achieve and maintain a healthy weight.

Nevertheless, the physical and psychological burdens induced by large amounts of PA and severe caloric restriction are beyond the perseverative capacity of most humans. As such, the long-term maintenance of weight loss becomes an increasingly unachievable goal as the structural and physiologic perturbations become more severe. Therefore, the management objective for individuals with inherited subtypes should be along the continuum of ‘fit but fat’. The refusal to appreciate this reality has led to unrealistic expectations, management ‘failure’, and the stigmatization of individuals with an inherited obese phenotype [71,72].

## 9. Assumptions and Limitations

Our ‘perspective’ is based on several assumptions that may limit our conclusions. The most critical is that obesity and cardiometabolic diseases are wholly anatomical (structural) and physiological (metabolic) disorders. Thus, we posit that if psychological, social, economic, or other non-physiologic phenomena influence obese or diabetic phenotypes, they must act through cellular mechanisms that cause increments in skeletal muscle-cell insulin-resistance and its sequelae (e.g., hyperphagia, adipogenic caloric partitioning, and increased fat-cell mass and number).

Although a large body of experimental evidence demonstrating the causal effects of PI on skeletal muscle-cell insulin resistance and its sequelae exists, the only support for speculations regarding the effects of psychological, social, and economic phenomena is correlational.

Moreover, we assert that distinguishing between etiology and treatment is critical for discussions revolving around the roles of PA, genes, diet, and exercise. For example, we contend that although specific macro-nutrients are not causal to obesity and other disease states, except as a source of calories (for details please see [69,73,74,75,76]), we argue strongly that caloric restriction with further reductions in carbohydrates are essential protocols for reducing body and fat mass and the treatment of acquired obesity and T2DM.

Finally, although our work on nongenetic inheritance and the developmental origins of disease is rigorous, consilient, and supported by voluminous research across species (please see [12] for details), our theories are novel and may therefore appear controversial to those unfamiliar with this emerging area of research and science (for reviews see [3,17,18,19]. Nevertheless, it remains to be seen if our conclusions withstand the ‘test of time’.

## 10. Summary and Conclusions

The age and rate at which an individual’s obese phenotype develops is a strong determinant of its response to intervention. Thus, the development of effective management strategies necessitates a personalized approach that incorporates the subtyping of obese phenotypes by etiologic status (acquired or inherited). The acquired phenotype denotes the development of excessive adiposity after puberty and is essentially a behavioral phenomenon induced by low levels of PA and concomitant hyperphagia (overconsumption). Thus, effective prevention and treatment strategies can be based on diet and exercise [32,47,77]. Although this subtype is amenable to lifestyle interventions in the early stages of development, the longer the PI, excessive SB, and overconsumption continue, the less amenable to intervention this subtype becomes.

In contrast, inherited obesity subsumes all forms of excessive adiposity that develop prior to pubescence (pediatric and childhood). The prevention of non-genetic inherited obese phenotypes in the next generation necessitates adequate levels of PA by the current generation of young females, potential mothers, and pregnant women. Nevertheless, once instantiated during the prenatal period, this subtype has irreversible structural (anatomic) and physiologic (metabolic) perturbations that are not amenable to intervention because no amount of diet and exercise can reduce the excessive number of fat cells and adipogenic partitioning, or significantly improve skeletal muscle function. Therefore, the objective in the management of inherited subtypes is the development of a ‘fit but fat’ phenotype. Importantly, because the amount of PA and caloric restriction necessary for the maintenance of weight loss with an inherited subtype may be beyond the physical and psychological capabilities of most individuals, it should not be the goal.

In closing, clinicians and investigators must recognize that despite its ubiquity, obesity is not a homogenous condition. Moreover, because obesity is a complex and idiosyncratic phenotype determined by inherited, behavioral, and environmental factors, a personalized approach based on etiologic subtype is essential for successful health management.

## Figures and Tables

**Figure 1 nutrients-14-02286-f001:**
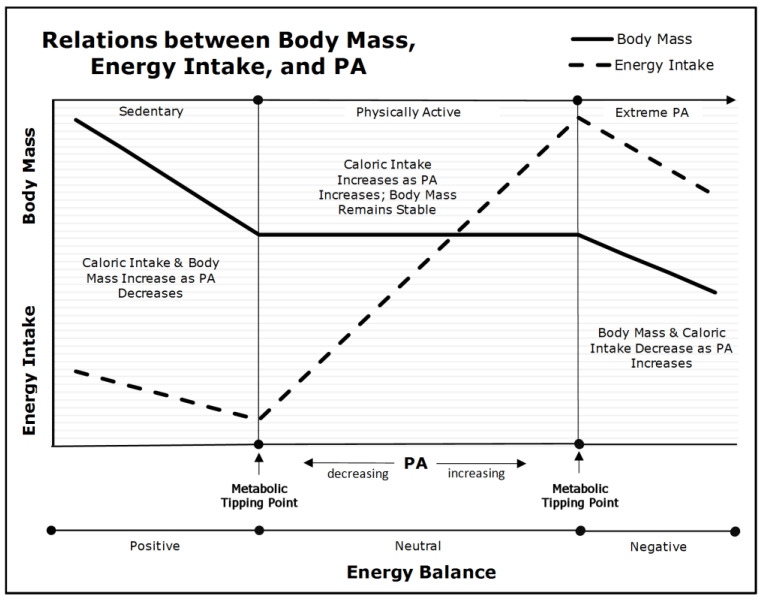
Relations between PA, Body Mass, and Energy Intake (adapted from [69]). As PA declines below the lower metabolic tipping-point into the ‘Sedentary’ range (left panel), energy intake and energy expenditure become dissociated due to insufficient PA. Body mass begins to increase as energy balance becomes positive and insulin sensitivity is diminished.

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
