# Peer review of "Obesity Subtyping: The Etiology, Prevention, and Management of Acquired versus Inherited Obese Phenotypes"

_nutrients, 2022, doi:10.3390/nu14112286_

Round 1

Reviewer 1 Report

In this perspective authors do repeat a series of statements supported by their own previous commentaries and perspectives with worrying lack of referencing the most updated literature on the aetiology of obesity and metabolic syndrome.

I would encourage the authors to have a careful inspection of the latest findings from the last 3 decades in both basic and clinical science. And use studies from the fifties and seventies to have a sense of how advances in all fields of science have allowed to better define, correct and in some cases debunk previous works on the understanding of human metabolism, specifically on the regulation of energy balance and their many contributing factors. And importantly, how there is still lots of things to understand and to discover.

Author Response

We thank the reviewer for the comments and critiques—and the opportunity to improve our manuscript.

To address the critiques, we added two strong sections. The first, titled “3. Nongenetic versus Genetic Inheritance and the Role of Genes in Obesity” presents conclusions based on early work in evo-devo biology and the emerging science of non-genetic inheritance (please see Lines 82-122 in the revised manuscript). We cited and quoted several of our prior papers as well as the work of other investigators. For example, see Lewontin (1974),  Feldman & Lewontin (1975) and for a broad overview, Bonduriansky R, Day T. Extended heredity: a new understanding of inheritance and evolution. Princeton, NJ: Princeton University Press; 2018.

It is important to note that our current article is a ‘Perspective’ and not a detailed review of other investigators’ research. Accordingly, in the revised manuscript we directed readers to a selection of our previous publications on nongenetic inheritance for greater detail and support.

The second additional section, titled,9. Assumptions and Limitations“ (please see Lines 333-353 in the revised manuscript) offers an improved context for understanding our perspective.

Once again, we thank the Reviewer for their insightful critiques and comments. Our paper is greatly improved as a result of these efforts and look forward to future discourse.

Sincerely,

Drs. Archer & Lavie

Reviewer 2 Report

Thanks for the opportunity to review this perspective review paper. This review provided a concise summarizing of obesity sub-phenotypes. Overall, the opinions are clear and the description are informative with sufficient references. However, in the manuscript, the understandings of the etiology of obesity are largely based on physiological models. However, obesity, either acquired or  inherited, could be influenced by lifestyle, nutrients, physiological status, sleep, and social-economic environment, in populations. The authors may need to expend a bit on their limitations that they did not discuss those aspects.
      Secondly, the tittle of this review mentioned "Prevention, and Management", however, little information was provided on the prevention and management strategies, instead, most of the manuscript are about physiological evidences. Modification of the title may be needed.

Author Response

We thank the reviewer for the comments and critiques—and the opportunity to improve our manuscript.

To address the critique of our “physiological models”, we added a strong section detailing the limitations of our hypotheses regarding psychological, social, and economic phenomena. Please see 9. Assumptions and Limitations“ (Lines 332-355 in the revised manuscript). We think that this new section offers an improved context by which to judge our ideas.

Moreover, we agree that both section 5 “The Prevention and Management of the Acquired Obese Phenotype “ and section 7 “The Prevention and Management of the Inherited Obese Phenotype” needed improvement. As such, we added greater detail with additional referential support. Please see Lines 161-164 (and new references 31, 42, 43) and Lines 242-244 in the revised manuscript.

Once again, we thank the Reviewer for their insightful critiques and comments. Our paper is greatly improved as a result of these efforts and look forward to future discourse.

Sincerely,

Drs. Archer & Lavie

Reviewer 3 Report

The manuscript titled "Obesity Subtyping: The Etiology, Prevention, and Management 2 of Acquired versus Inherited Obese Phenotypes", is an interesting  review shedding light on the subtype-specific obesity prevention and management. The field is very fascinating, the paper is well written. The results are clearly reported and well discussed. 

Author Response

We thank the reviewer for these comments and compliments. We too find the field “very fascinating” and appreciate the effort made to review our work.

Sincerely,

Drs. Archer & Lavie